# Photo Composition with Real-Time Rating

**DOI:** 10.3390/s20030582

**Published:** 2020-01-21

**Authors:** Yi-Feng Li, Chuan-Kai Yang, Yi-Zhen Chang

**Affiliations:** 1Department of Information Management, National Taiwan University of Science and Technology, Taipei 106, Taiwan; yifengli@gmail.com; 2CyberLink, New Taipei City 231, Taiwan; taco840602@gmail.com

**Keywords:** photo composition, real-time rating, GrabCut

## Abstract

Taking a photo has become a part of our daily life. With the powerfulness and convenience of a smartphone, capturing what we see may have never been easier. However, taking good photos may not be always easy or intuitive for everyone. As numerous studies have shown that photo composition plays a very important role in making a good photo, in this study, we, therefore, propose to develop a photo-taking app to give a real-time suggestion through a scoring mechanism to guide a user to take a good photo. Due to the concern of real-time performance, only eight commonly used composition rules are adopted in our system, and several detailed evaluations have been conducted to prove the effectiveness of our system.

## 1. Introduction

Due to the extensive usage of smartphones and the popularity of social media, a lot of people use camera phones, smartphones with camera functions, to take photos in various ways on different occasions. While taking a photo, people tend to arrange elements within it in a way that suits best the goal of their work, or just leaves an impression of what they see.

Photo composition is a way to express a photographer’s thoughts with the processes of how he/she constructs the image and is a key element of visual effects. With proper composition techniques, photographers can guide viewers’ attention to the main subjects of photos. Although aesthetical judgement could be very subjective or personal, several objective composition rules have been proposed by experts so that novices can learn to construct good images more efficiently, such as rule of thirds, horizontal composition, etc.

Currently, most researches propose some image aesthetical analysis after an image is taken. In this paper, we propose a smartphone application that can evaluate images instantly while the images are being taken. Our application allows users to choose desired composition methods and provides corresponding evaluation scores continuously and instantly for each image while the application is running. As a result, during the photographing process of using our application, users can learn how to take photos according to some commonly known composition rules. Figure 1 is an example of our application interface, which contains information including the current selection and parameter settings of involved composition rules and the resulting score of the current scene.

To sum up, the contribution of this paper is twofold. First, we have developed a system that could provide a real-time scoring APP based on some commonly used composition rules to efficiently guide a user during the photo-taking process. Second, we also allow a user to select or customize the involved composition rules to not only suit the user’s personal needs, but also to facilitate the photo taking process because once the specified rules/conditions are met, the photo will be automatically taken and stored. We have conducted various evaluations to further validate the effectiveness of our system and an experiment with/without using our system to demonstrate that such a system can indeed assist users to take “better” photos.

The rest of this paper is organized as the following. Section 2 reviews some studies that are more related to this work. Section 3 details how we make use of several composition rules to rate a photo. Section 4 demonstrates some results generated by our system. Section 5 compares our results with the ones taken by various photographers when applicable and also by a random taking process to show the effectiveness of using our system. Section 6 concludes this paper and also hints for potential future directions.

## 2. Related Work

There have been numerous books describing some general rules to take good photos, and among them, Grill et al. [1] has published a book that once becomes one of the most popular books on photography. These rules are normally called composition methods or composition rules. In addition to the classic line and shape composition, there are also composition rules based on light and color. We believe that it is not absolutely right or wrong whether one of such rules should be applied or not, as the goal really depends on how the user wants to capture an image to convey some artistic concept or simply to preserve an impression. We could say that these rules may be helpful for an amateur; however, it may not always be easy to check against so many rules at one time, let alone to memorize so many rules could be a burden to most people.

To address this, Yeh et al. [2] proposed a novel personalized ranking system for amateurs with two different interfaces. The goal is to help a user rank a collection of photos that have been taken so that it’s easy to determine whether some of the photos should just be discarded. One interface is feature-based to allow a user to adjust feature weights to rank photos. The other interface is example-based that the system makes use of an example images provided by a user to rank photos. They proposed to use rules of aesthetics in three categories: photograph composition, color and intensity distribution, and personalized features. Photograph composition includes rule of thirds, size of region of interest (ROI) and simplicity features. Color and intensity distribution includes texture, clarity, color harmonization, intensity balance, and contrast. Personalized features include color preference, black-and-white ration, portrait with face detection and aspect ratio. With the adopting of ListNet [3] for the trained ranking algorithm and retrieving features weightings set by users, the system generates a feature-based ranking list to rank photos. On the other hand, example-based ranked images are ordered with the weighing vector derived based on the example photographs. They later on proposed to refine the photo ranking system for amateurs by an interactive personalization approach [4] so that each person gets to define his or her own ranking through positive and negative feedbacks. Carballal et al. [5], with the target being landscape images, proposed some features and metrics that could be used for estimating the complexity of an image and a binary classifier is trained through the help of an expert to judge an image’s aesthetic quality with the the accuracy of 85%. Obrador et al. [6] proposed some low-level image features such as the simplicity of a scene, visual balance, golden mean and golden triangles and managed to achieve comparable aesthetic judgement, compared with the start-of-the art of their times. Deng et al. [7] surveyed image aesthetic assessment methods with the aim of being a binary classifier, i.e., can tell if an image is good or bad, with the particular mentioning of some recent deep-learning approaches. Wu et al. [8] proposed to use a deep neural network or DNN for short to consider 12 composition rules, including several novel ones that have not been studied before, and the results were evaluated by Amazon Mechanical Turk (AMT) to prove the effectiveness of their approach. On evaluating the composition rules, Luo et al. [9] proposed to focus more on the foreground subjects, and provided some examples to demonstrate their claims. The evaluation is also extended to videos by considering motion stability, which also plays an important role on the resulting video quality. Abdullah et al. [10] also made use of composition rules, but in a rather different context, to help place a virtual camera in a computer graphics rendering scenes that satisfy some common or important composition criteria.

While most people agree that composition rules are important for taking a photo, it would be nice to have an automatic and real-time evaluation of the current composition to determine if such a photo should be taken or not. Regarding this, Xu et al. [11] proposed a real-time guidance camera interface; however, only rule of thirds is considered. Anon [12] filed a patent for real-time composition feedback defining some generally desired hardware and software specifications without elaborating on how to achieve real-time performance if many composition rules are involved. In fact, as far as we know, most works are for the evaluation or ranking afterwards, i.e., after the photos have been taken. On the contrary, we would like to give a real-time suggestion/response through a scoring mechanism to hint or instruct a user about how to take a good photo by considering many commonly used composition rules. In particular, due to the constraints of resource and real-time performance, currently, only eight composition rules are implemented in our proposed system, and will be detailed in the next section.

## 3. Composition Analysis

### 3.1. System Overview

As shown in Figure 2, after launching our APP which will initiate the camera function at the same time, users are allowed to select the desired composition methods; then the APP will analyze each image frame and provide a corresponding score immediately based on the preferred techniques. Note that the input is called an image frame is because it is in fact decoded from the captured video when the camera mode is turned on. Each composition method involves different key techniques. The horizontal composition is to detect the longest line in an image. The rule of thirds is to find the center of gravity of the largest saliency area within an image. The triangle composition discovers the smallest triangle which can enclose the largest saliency region. The vanishing point technique is to search the intersection point with the highest number of lines. The frame within a frame technique calculates a rectangular area of the background. With Sobel operation and integral image, the focus composition rule finds the clearest area. The intensity balance technique compares the difference of gray scale value between the right-half image and the left-half image. Using the V channel of HSV color space, the contrast method calculates the difference between all pixels and the average pixel value. Each composition method generates a corresponding score. The APP will provide the average score of desired methods as a reference for the user to evaluate the quality of the image frame.

### 3.2. Composition Analysis

In this section, we will discuss the implementation of each composition rule and how we calculate the corresponding scores.

#### 3.2.1. Horizontal Composition

Horizontal composition tends to provide a feeling of stability and calmness to the image viewer because a lot of scenes are horizontal in nature [1,13]. For instance, Figure 3 is an image of a beach with a horizon parallel to the ground and it gives a sense of calmness to the viewer.

In order to implement the horizontal composition method, we perform the following steps. First, we transfer the image to gray scales and apply Gaussian blur to eliminate the potential image noise. Second, we do edge detection by adopting the proposed method from Canny [14]. Figure 4b is the edge detection result of Figure 4a. Third, we use Hough Transform [15] with the setting of parameters to allowing two segmented lines being merged to one straight line to detect all possible lines. In the meanwhile, a filter is applied to filter out the lines with θ greater than 30 degree, where θ is the angle between the detected longest line and the referenced horizontal line. Finally, as shown in Equation (Equation 1), we use the θ of the detected longest line to calculate the resulting score of horizontal composition, $S_{horizon}$. If θ is 0, it will lead to the highest score. Figure 4c demonstrates the θ of the detected longest line of the example image.
(1)$$S_{horizon}=100-\left(\theta /30\right)*100$$

#### 3.2.2. Rule of Thirds

The rule of thirds is the most well-known and applied image composition guideline [1,2,4,6,7,8,9,10,11,13,16,17,18]. As illustrated on Figure 5, the image frame is divided into nine equal rectangles with two vertical gridlines and two horizontal gridlines. The theory of the rule of thirds is to make the image visually pleasing by placing the (foreground) subjects on or near the gridlines or the intersection (red dots) of the gridlines. With proper spacial arrangement, this technique will help the user to create a more balanced photo and to provide restful feeling for image viewers.

Yeh et al. [2] proposed a method to evaluate if the image conforms to the rule of thirds properly or not. Basically, this method applies the segmentation technique and the saliency values in the Lab color space to retrieve the main subjects of the image. Then, it evaluates the effect of the rule of thirds using the distance between the main subjects and the intersections of gridlines. Note that the closer the main subject to the gridline intersections, the better the score. Figure 6 is a composition example of rule of thirds.

The most important task of the rule of thirds is to identify the main object within the image. In this paper, we use the image saliency map to retrieve the main object. Although Rother et al. [19] proposed a method, GrabCut, which may get a better result, we adopt Gildenblat proposed method [20], due to the concern of execution time.

More specifically, we adopt the method of utilizing histogram back-projection to simplify the detection of a saliency map, inspired by Gildenblat [20]. First, the Gaussian Blur is applied to remove image noise. Second, the image is transferred to HSV color space. With the normalization of the range from 0 to 255, the histogram value is calculated under H and S color channels. Finally, a back-projection image is retrieved from back-projecting H and S channel histograms. In the back projection, we calculate the value of a pixel, divided by how many pixels are with the same value, so that each image pixel value in the back-projection image stands for the background probability of the corresponding pixel. A darker color means that the corresponding pixel has a higher probability to present a foreground object. Figure 7b is an back-projecting example of Figure 7a.

After getting the back-projecting image, we reverse the image, as shown in Figure 7c, so that the whiter color presents the foreground object. Then, the mean shift [21] is adopted to make smooth color details and erode small color regions, as shown in Figure 7d. Also, histogram equalization is applied to enhance the image contrast, shown in Figure 7e. With an image binarization, we get the saliency map of the image, shown in Figure 7f.

After retrieving the saliency map, we adopted the method proposed by Suzuki et al. [22] to get the largest salient region as the main object of the image by calculating all borders of salient objects. Followed by the calculation of characteristic moment, we can get the corresponding rectangle of the main object and its center of gravity. In Figure 8, the blue rectangle indicates the outline of the main object and the central blue point indicates its center of gravity.

The position of the main object’s center of gravity is used to evaluate the score for the rule of thirds. When the position is closer to one of the six red dots, shown in Figure 9, it will get a higher score. We use Euclidean Distance, as was used by Yeh et al. [2], to get the shortest distance, shown in Equation (Equation 2) where $M_{x}$ and $M_{y}$ are the x-axis and y-axis values of the center of gravity of the main object. Note that sometimes a user may prefer to use central composition, as opposed to rule of thirds, and in that case, the distance will be calculated between the position of the main object’s center of gravity and one of the green dots, shown in Figure 9.
(2)$$distance=\sqrt{{\left(M_{x}-point.x\right)}^{2}+{\left(M_{y}-point.y\right)}^{2}}$$

After getting the shortest distance, we use Equation (Equation 3) to calculate the score for the rule of thirds: $S_{rot}$. Here we normalize degree to be in the range of 0 to 90 degree, and we can use the function of radians to convert from degrees to radians. Also note here that worst is the largest possible distance, namely the distance between the leftmost (rightmost) corner position and the leftmost (rightmost) red/green dots.
(3)$$\left\{\begin{array}{cc} degree & =90\times \frac{1}{worst}\times distance\\ S_{Rot} & =cos(Radians(degree))\times 100 \end{array}\right.$$

#### 3.2.3. Triangle Composition

Triangles are generally regarded as stable shapes and are often used to denote balance [16,17]. Triangle composition is that the main subjects should portray the shape of a triangle so that it could bring stability, peace, and harmony to the image [23]. An example is shown in Figure 10.

As discussed previously in the rule of thirds case, we can find the salient region of the main object and then use Ahn et al. [24] proposed method to find the smallest triangle to enclose the salient region. For example, as shown in Figure 11, the red triangle is the smallest triangle, including the largest salient region of Figure 7f.

We evaluate the score of triangle composition based on the related angles of the smallest triangle including the main object. Based on the principle and the description from [23], an isosceles triangle will get the highest score. Equation (Equation 4) is used to calculate the angle, where A, B, and C represent the corner points, and $\bar{AB}$, $\bar{AC}$, and $\bar{BC}$ are the distance between two points.
(4)$$\left\{\begin{array}{cc} cosA & =\frac{{\bar{AB}}^{2}+{\bar{AC}}^{2}-{\bar{BC}}^{2}}{2\times \bar{AB}\times \bar{AC}}\\ angleA & =arccos(cosA)\times 180/\pi \end{array}\right.$$

After getting all three angles of the triangle, we use Equation (Equation 5) to evaluate the score of triangle composition, where θ is the smallest difference between any two angles and the θ of 59 degrees can be shown is the worst case after it is rounding to an integer.
(5)$$S_{triangle}=100-\left(\theta /59\right)\times 100.$$

#### 3.2.4. Vanishing Point

The vanishing point composition is one of the best tricks for creating the impression of depth in landscape images [13,16,17,18,25]. The theory behind vanishing points is closely related to perspective viewing where parallel lines seem to converge as they get progressively further away. This technique can provide a strong visual cue to attract more attention from image viewers. In realty, the phenomenon of vanishing points can be found in many cases, such as streets, rivers, and railroads, etc. Figure 12 is an example of vanishing point composition.

For the analysis of vanishing points, we detect all straight lines within an image by using the method discussed previously (horizontal composition), and then filter out horizontal and vertical lines. After the filtering, we use Equation (Equation 6) to represent the line equation of each line, where ($x_{1}$, $y_{1}$) and ($x_{2}$, $y_{2}$) are the endpoints of a straight line.
(6)$$\left\{\begin{array}{cc} a & =-(y_{2}-y_{1})\\ b & =x_{2}-x_{1}\\ c & =ax_{1}+by_{1}\\ ax+by & =c \end{array}\right.$$

Through linear algebra operation of all line equations, all intersection points, ($x_{0}$, $y_{0}$), of any two lines can be calculated. Then, using Equation (Equation 7), the minimum distance, minDistance, can be calculated between the intersection point and any other lines, ax+by=c.
(7)$$minDistance=\frac{|ax_{0}+by_{0}+c|}{\sqrt{a^{2}+b^{2}}}.$$

In the end, we sum up all minimum distances from an intersection point to all lines and treat the point that has the minimum accumulated value as the vanishing point of the image. Figure 13b is the edge detection result for Figure 13a. In Figure 13c, all red lines are detected as straight lines after filtering, and the red dot is the vanishing point.

The position of the vanishing point will affect the evaluation result. We use Figure 14 as a reference, if the vanishing point is closer to one of the seven red dots shown in the figure, it will get a higher evaluation score, $S_{vanishing}$. Equation (Equation 8) shows how we calculate the score, where $worst_{d}$ illustrates the worst distance between any point to one of the seven reference points.
(8)$$\left\{\begin{array}{cc} worst_{d} & =\sqrt{{\left(width/3\right)}^{2}+{\left(height/3\right)}^{2}}\\ S_{vanishing} & =100-(distance/worst_{d})\times 100. \end{array}\right.$$

#### 3.2.5. Frame within a Frame

Frame within the frame is another effective method of portraying depth in a scene [8,13,16,17,18]. It involves using objects in the foreground of a scene to create a frame around the main subject. Archways, doorways, windows, tree branches and holes could make perfect frames. Placing these kinds of objects around the edge helps isolate the main subject, drawing our attention towards it. As well as creating more visual interests, a frame adds some meaning to a picture as it puts the main subject in context with its surroundings. Figure 15 is such an example.

In order to evaluate frame within a frame, this paper uses GrabCut, proposed by Rother et al. [19], to isolate the foreground and background of an image and to form an image mask. Then, we try to get the background image with a maximum border by adopting Suzuki et al. proposed method [22]. Other regions outside the border are treated as the foreground image frame. As shown in Figure 16, Figure 16a is an original image; Figure 16b is the mask image; Figure 16c is the background image; Figure 16d is the foreground image frame.

We use the area of the extracted background image to evaluate the score for the effect of frame-within-a-frame composition. An initial rectangular area is set for the GrabCut to perform the background extraction for two reasons. First, it’s easier to automatically specify than other types of area. Second, it contains a larger interior area because its boundary is closer to the image boundary. If the resulting area from the GrabCut is closer to the initial rectangular area, the score is higher. More specifically, the evaluation score, $S_{frame}$, is calculated according to Equation (Equation 9), where area is the area of the background region, while best and worst are the parameters of the best case and the worst case, respectively. For the adjustment details of best and worst, please refer to Section 4.2.
(9)$$S_{frame}=area\times \frac{100}{best-worst}-\frac{100\times worst}{best-worst}.$$

#### 3.2.6. Focus

Focus composition is a starting point of creative and composition [1,9,13]. It helps the viewers to identify the main subjects. Each image should emphasize and expand from the main subject. The focus setup is a very important procedure when constructing an image.

For focus composition analysis, we have to identify the clearest region first. We transfer the image to grayscale by using the Sobel operator to calculate vertical and horizontal gradients separately, and then transform the gradient values into grayscale values. Furthermore, we use the weighted average of these two gradients to get the edge detection image. Figure 17b is the edge detection result of Figure 17a.

After edge detection with the Sobel operator, this paper adopts the idea of integral images [26] to speed up the involved computation to identify the clearest region based on the largest total pixel value within the block. We use Equation (Equation 10) to perform operations on the integral images of Sobel results and Figure 18a illustrates the operations.
(10)$$Integral\left(x,y\right)=\left\{\begin{array}{ccc} = & sobel(x,y) & if\;\;x=0\;\;and\;\;y=0\\ = & Integral(x,y-1)+sobel(x,y) & if\;\;x=0\\ = & Integral(x-1,y)+sobel(x,y) & if\;\;y=0\\ = & Integral(x-1,y)+Integral(x,y-1) & otherwise\\ \; & -Integral(x-1,y-1)+sobel(x,y) \end{array}\right.$$

We then use Equation (Equation 11) to calculate the total pixel value of a rect(angular) block, where the block size is (b × b) and the starting point is (x-b, y-b). Figure 18b illustrates the operations.
(11)rect=Integral(x,y)−Integral(x−b,y)−Integral(x,y−b)+Integral(x−b,y−b).

The yellow rectangle near the center of Figure 17a indicates the clearest region because of its largest total pixel value.

There are two parts to evaluate focus composition and each part has the score ranging from 0 to 50. The first part is based on the clear level of the clearest region and the score, $S_{focus1}$, is calculated by Equation (Equation 12), where best and worst are the adjustable parameters whose adjustment details are discussed in Section 4.2.
(12)$$\left\{\begin{array}{cc} S & =\frac{rect}{b\times b\times 255}\times 50\\ S_{focus1} & =S\times \frac{100}{best-worst}-\frac{100\times worst}{best-worst}. \end{array}\right.$$

The second part is based on the Euclidean distance (also used here to be consistent with rule-of-thirds judgement related to the position of the main object), denoted as distance in Equation (Equation 13), between the central point of the clearest region and the main object’s center of gravity. The evaluation score, $S_{focus2}$, is calculated by Equation (Equation 13). We assume that the distance less than or equal to b/2 can achieve the best result and that the longest distance, maxdis, between the main object’s center of gravity and the furthermost corner of the image, minus b/2 is the worst case.
(13)$$S_{focus2}=50-\frac{distance-b/2}{maxdis-b/2}\times 50.$$

It can be seen that highest score for each part is 50, so $S_{focus1}$ and $S_{focus2}$ are added up as the evaluation score, $S_{focus}$, for focus composition.

#### 3.2.7. Intensity Balance

Balance can provide the feeling of calmness and is a very basic element of visual effect [2,4,8]. Yeh et al. [2] proposed a method of using an image histogram to evaluate the intensity balance, based on the chi-square distribution difference of the left-half and right-half of an image. Figure 19a,b demonstrate a balanced and un un-balanced images.

With Yeh et al. proposed method [2], we compare the histogram of the left-half image and the right-half image to evaluate the intensity balance of an image. Figure 20b and Figure 20c are the left-half histogram and the right-half histogram of Figure 20a.

As described in Equation (Equation 14), we use the difference, $diff_{H}$, between the left-half histogram ($H_{left}$) and the right-half histogram ($H_{right}$) to get the intensity balance evaluation score, $S_{intensity}$, where *w* and *h* are the image width and height, respectively. The worst difference, worst, can be presumed that there is no common histogram value from both halves.
(14)$$\left\{\begin{array}{cc} diff_{H} & =\sum_{x=0}^{255}\left(\right|H_{right}\left[x\right]-H_{left}\left[x\right]\left|\right)\\ worst & =w\times h\\ S_{intensity} & =100-(diff_{H}/worst)\times 100. \end{array}\right.$$

#### 3.2.8. Contrast

Contrast means comparison and difference [1,2,4,7,9,13,18]. Contrast can assist a photographer to create eye-catching images with visual layers and can direct viewers’ attention to the main subjects. For example, a dark background is usually used to emphasize the bright subjects. Yeh et. al. [2] proposed two methods to evaluate contrast. One method uses Weber contrast to calculate the average intensity difference of all pixels with respect to the average intensity of the image. The other method is to use CIEDE2000 color difference equation [27] to calculate color contrast. Figure 21 are examples of images with different degree of contrast, where Figure 21a has high contrast and Figure 21b has low contrast.

With Yeh et al. method [2], we transform the image to the HSV color space and evaluate the contrast performance based on the histogram of the V channel, $hist_{V}$. As shown in Equation (Equation 15), we count the average pixel value of the V channel, $average_{V}$, where *w* and *h* are the width and height of the image, respectively.
(15)$$average_{V}=\sum_{x=0}^{255}\frac{\left(hist_{V}\left[x\right]\times x\right)}{w\times h}.$$

We then calculate the total difference, $diff_{v}$, between each image pixel value and the $average_{V}$ using Equation (Equation 16).
(16)$$diff_{V}=\sum_{x=0}^{255}(|x-averageV|\times hist_{V}\left[x\right]).$$

In the end, we calculate the contrast score, $S_{contrast}$, as shown in Equation (Equation 17). We set the score to be 100, if calculated result is greater than 100.
(17)$$S_{contrast}=\frac{diff_{V}}{w\times h}.$$

### 3.3. Weighted Combination

We use the weighted average score as the final evaluation score. Currently, we set the weight as 1 for each selected composition method, and 0 for those that are not selected. Our system considers at most eight different composition methods. Equation (Equation 18) shows how the final evaluation score is calculated, where s and w represent scores and weights, while N is the number of selected methods, respectively.
(18)$$Score=\sum_{x=1}^{8}\left(s\left[x\right]\times w\left[x\right]\right)/N.$$

### 3.4. Acceleration

#### 3.4.1. Multi-Threading

In order to provide analytical results instantly, the proposed system adopts a multi-thread processing and evaluates one frame out of every 10 frames due to limited resources on a mobile device like a smartphone.

#### 3.4.2. Downsampling

A high-resolution image usually requires more processing time. Even with a multi-thread processing, we found out that the result analysis would be delayed for processing images with its original resolution of 960 × 720. Thus, a pre-processing of down-sizing images is performed before evaluating each composition method. For frame within a frame, which requires more complicated operations, we down-size the image resolution five times to become 192 × 144. In Table 1, we can see that the operation time is reduced from 5.658 s to 0.073 s. For other composition techniques, we down-size the image resolution three times to become 320 × 240. For focus composition, the technique of image pyramid is also adopted to down-size the Sobel edge detection image 12 times to become 80 × 60 to expedite the process of finding the clearest region. Then, the image is up-sized 4 times to 320 × 240 for further processing. More system environment details are discussed in Section 4.1. Note that, although an image down-sampling process may lose the image details, the impacts are minor for most our composition evaluation techniques. For example, we down-sample the image to a reasonable size with the same aspect ratio on its width and height to maintain the basic geometric image features accordingly and properly, such as the straight lines, the main subject’s location and it’s bounding box. Also, a linear-averaging of the corresponding pixels are applied in the down-sampling process so that the impacts are small color-wisely for color-related techniques such as focus, intensity balance, and contrast composition techniques. To further confirm this, we had evaluated 50 images before and after down-sampling and the corresponding scores are very close, that is, with less than 5% difference.

#### 3.4.3. Saliency Extraction

In Section 3.2.2 (rule of thirds), a saliency map detection is required for evaluating rule of thirds. Under the comparison of processing images with the resolution of 320 × 240, the performance of GrabCut and Gildenblat proposed method [20] are 0.03 s and 0.207 s, respectively. Gildenblat’s method improves the performance remarkably.

#### 3.4.4. Image Integration

In Section 3.2.6 (focus), finding the clearest region is required for focus composition. Under the image resolution of 320 × 240, it takes 0.726 s for a normal three-layer loop search. However, the performance can be expedited to 0.04 s with the use of integral images on Sobel edge detection results.

#### 3.4.5. Histogram Computation

In Section 3.2.6 (focus), the average pixel value difference under the V channel is required for contrast composition. With a histogram pre-processing, we enhance the evaluating efficiency by using only two single-loop computations, rather than two two-layer loop computations, which generally require more time.

## 4. Results

### 4.1. System Environment

The system environment is listed in Table 2.

### 4.2. Parameter Setting

In our implementation, we adopt the OpenCV built-in library. Most parameter settings are limited and described in the library. Basically all other parameters are determined in a ‘trial and-error’ manner to strike a reasonable balance between accuracy and performance. In particular, in Section 3.2.1 (horizontal composition), there are two thresholds for Canny edge detection; we set threshold1 as 50 and threshhold2 as 100. For Hough transform to detect straight lines, we set the accumulation threshold (threshold) as 10, minimum distance (minLineLength) as 10, and the maximum gap (maxLineGap) as 2, respectively.

In Section 3.2.4 (vanishing point), we set threshold1 as 70 and threshold2 as 120 for Canny edge detection. For Hough transform straight line detection, we set threshold as 10, minLineLength as 10, and maxLineGap as 0, respectively.

In Section 3.2.2 (rule of thirds), the parameter, worst, means the longest distance that we expected for the evaluation score to be 0, between the main object’s center of gravity and one of the intersection points. Based on two experimental results, taking photos with and without considering system’s suggestions for images under 320 × 240 resolution, we set worst to be 100 because there are hardly images with the relevant distance longer than 100. Figure 22a,b show the experimental results of the relation between the number of images and the relevant distances.

Note that the choice of worst may change according to different image resolutions, we, therefore, adjust its value based on one-third of the diagonal length of an image. Assume *x* is one-third of the diagonal length of an image, and $\bar{x}$ is one-third of the new image’s diagonal length, then the worst value should be adjusted as $\frac{\bar{x}}{x}\times 100$.

In Section 3.2.6 (focus composition), we also conduct two experiments for distinguishing clarity levels, and set the scores in the range from 0 to 50. In Figure 23a,b, we can see that the scores are in the range from 12 to 40 for both experiments, i.e., taking photos with and without system’s instructions. Thus, in Equation (Equation 12), we set best to 40, and worst as 12, respectively.

In Section 3.2.5 (frame within a frame), GrabCut function is used to separate the foreground and background from an image. According to the observation of experimental results under image resolution of 192 × 144, for input rectangles (rect), we set the parameters as follows: width as ($Image_{width}$: 30), height as ($Image_{height}$: 20), and intiterCount as 1, respectively, so that foreground and background images can be identified more accurately. As a result, in Equation (Equation 9), we assume the parameters: worst as 30 × 20, and best as ($Image_{width}$: 30) × ($Image_{height}$: 20), respectively. With different sizes of down-scaled images, the proportional adjustment will be done accordingly. Figure 24 are examples with a frame-within-a-frame composition. Figure 24a is 1200 × 1600; Figure 24b is 1200 × 1600; Figure 24c is 3264 × 2448; Figure 24d is 1200 × 1600. While Figure 24e–h are the corresponding foreground results after applying the GrabCut.

### 4.3. System Interface

Figure 25a is an example of the application main screen. There are 3 functional buttons on the right side. A user can press the central circular button to take a photo. The number at the bottom is the composition score evaluated by the system. With a light touch on the score, a bar chart will be displayed to indicate each composition score, shown in Figure 25b. If we press the button at the top, the system will allow users to select single or multiple desired composition methods, shown in Figure 25c. Note that either central composition or rule of thirds, but not both, can be selected. Also note that the relative weights for intensity balance and contrast can be adjusted through the interface.

On the composition selection interface, a user can long-press on any composition text for 3 seconds and a corresponding brief composition description will be displayed, as shown in Figure 26a. While the ‘Auto’ function is selected, the application will pop-up a screen for a user to fill in a target score value, as shown in Figure 26b. When the evaluation score is greater than or equal to the input score, the system will automatically take a picture and change to the interface as shown in Figure 26c, so that a user can press the central button to save the image or press the back arrow sign on the top to skip this option.

### 4.4. Results

In this section, with the selection of various composition techniques, we demonstrate and analyze the result images and their corresponding composition details accordingly.

Figure 27 is our first testing image and Table 3 shows its selected composition methods and their corresponding evaluation scores.

In general, for horizontal composition, we detected and displayed the longest reference line, mentioned in Section 3.2.1, in the image. For rule-of-thirds composition and central composition, we indicated the bounding box and the center of gravity of the detected main object along with the nearest reference point, mentioned in Section 3.2.2. For triangle composition, we displayed the smallest triangle that includes the main object, as mentioned in Section 3.2.3. For focus composition, we show the clearest region, as mentioned in Section 3.2.6. For vanishing point composition, we display detected straight lines after filtering out horizontal and vertical lines and indicate the calculated vanishing point and reference points, as mention in Section 3.2.4.

In particular, Figure 28 shows the detailed testing results for different composition rules based on our techniques described previously. Note that the scores for horizontal and rule-of-thirds are very high because of the long detected horizontal line in Figure 28a and the center (of gravity) of the main object is very close to one of the best reference points for rule-of-thirds, as shown in Figure 28b.

Figure 29 is our second testing image and Table 4 shows its selected composition methods and their corresponding evaluation score.

Figure 30 shows detailed testing results. Note that the reason for the focus score to be not high is due to the fact that the detected focal point, marked in yellow in Figure 30d, potentially affected by the lighting from the above, has a large distance to the main object’s center, thus causing the low score.

Figure 31 is our third testing image and Table 5 shows its selected composition methods and their corresponding evaluation scores.

Figure 32 shows the detailed testing results. One thing to observe from the original image in Figure 31 is that, in terms of a frame, the left, top and right regions altogether form roughly a frame, but the bottom part of the image is open, thus making the resulting frame-within-frame background region to be noticeably smaller than its initial area, as shown in Figure 32c, thus causing relatively lower values compared with other scores.

In order to verify our system’s evaluation scores, we use the application with various composition methods to take images for the same scene. In Figure 33, horizontal, rule of thirds, triangle, and focus compositions are applied, while in Figure 34, horizontal, vanishing point, frame within a frame, and focus composition are applied.

Note that We do not show the examples with selecting all rules because some rules are not compatible with each other. For example, the rule-of-thirds is not suitable to be with the frame-within-a-frame composition because the best position to consider for rule-of-thirds is affected indefinitely by the detected frame along the image boundary. In addition, the foreground frame could also affect the detection of the main object.

### 4.5. Limitations

As described in Section 3.2.4, we use straight line segments to find the vanishing points. However, this technique works only on the images with structural roads, such as the case shown in Figure 35a, and it cannot work precisely for non-structural roads, such as the case in Figure 35b.

Although Lu et al. [25] proposed to use multi-population genetic algorithm to detect various vanishing points, it requires much longer operation time, and as a result, we did not adopt this method.

For frame within a frame, our technique is based on GrabCut which works better for images with closed frames than open frames for distinguishing foreground and background from an image. In Figure 36, the foreground is isolated reasonably. In Figure 37, our system can not find a good foreground because this image does not have a perfect closed frame.

Besides, we observed that the performance of GrabCut is greatly affected by the image resolution. As shown in Figure 38 and Figure 39, GrabCut works better on the original images than on the down-scaled images.

## 5. Evaluation

### 5.1. Score Distribution

In order to observe the difference of evaluation scores between taking photos with and without using our application, and for the comparison purpose, we used three different image sets; (1) we download 9171 images taken by various photographers from the Internet [28]; (2) we take 324 images using our application; (3) we took 317 images without using our application, but in a random manner. More specifically, we used the smartphone to slowly cycle around our body to take a picture (simulating what we do when looking for a better photo composition) for every few frames until we have obtained a certain number of images. We then used our system to evaluate each image and get the corresponding scores for various composition methods. The histogram results are shown in the following sub-sections. Note that the X-axis represents the evaluation score while the Y-axis represents the corresponding number of images.

#### 5.1.1. Experiment Environment

We set the following evaluation environment: (1) Except for the case of the frame within a frame, we evaluate images at down-scaled resolution 320 × 240 or similar sizes for the downloaded images due to their various sizes. (2) For the case of the frame within a frame, due to the lack of sample images, we analyze only 140 downloaded images and 317 (randomly taken) images without using our application. All images are down-scaled to 192 × 144 or similar sizes. (3) We take photographs in 10 different scenes with and without using our application. (4) While taking photos using our application, we put the main objects around the center first, and then move the main objects to the intersection points mentioned in the rule of thirds. While taking photos without using our application, we also put the main objects in the center first, but then move main objects to any other places randomly.

#### 5.1.2. Experiment Analysis

According to the evaluation results for 8 different composition methods, score distributions of taking photos using our application are similar to those of the downloaded images, taken by various photographers. In addition, score distributions are more concentrated for images taken using our application than the ones without using our application, especially for the high score portion. This is not true for the photos taken (randomly) without using our application. The evaluation results are demonstrated as follows:

(1) Figure 40 are the results for horizontal composition. Note that, as the image counts for the score of 100 are too many, they are not shown in this figure, and their corresponding numbers are: (a) 2078 (out of 9171), (b) 111 (out of 324) and (c) 49 (out of 317) images.

Figure 41 is for the rule of thirds composition.

Figure 42 is for triangle composition.

Figure 43 are the results for vanishing point. Note that the image counts for score of 0, for being too many, are not shown in the figure, and their corresponding image numbers are: (a) 7062 (out of 9171), (b) 244 (out of 324), and (c) 255 (out of 317).

Figure 44 is for the focus composition. According to these results, there are similar distribution peaks for downloaded images and images taken using our application, but not the images (randomly taken) without using our application.

(2) There are not many images with frame within a frame so we compare 140 downloaded images, using the frame-within-a-frame composition and images taken without using our application. The results are shown in Figure 45. Note that we again filtered out images with score of 0, and the corresponding image numbers are: (a) 8 (out of 140) and (b) 160 (out of 317). According to the results, we can see that image distributions behave differently, where more images are with higher scores when the frame-within-a-frame composition rule is used, while more images are with lower scores when images are taken randomly.

(3) In Figure 46, we can see that the distribution of taking images using our application is more scattered. For getting high scores on intensity balance, as mentioned in Section 3.2.7, it requires symmetrical intensity histogram distribution between the right-half and left-half images. In Figure 46c, we can see some images with very high scores, which is due to the fact that some over-exposed images are included during the random photo-taking process. As a result, we set a default value of 90 for intensity balanced composition.

(4) In the contrast evaluation results, shown in Figure 47, we can see that (a) and (b) are more concentrated than (c) and the majority of image contrast evaluation distribution scores are in the range from 40 to 65. To have a good contrast score, the image should contain some clearly dark and bright regions, as shown in Figure 48. Thus, in the application, we set a default value of 50 for contrast evaluation.

### 5.2. Result Preference

In order to evaluate if our scoring methods fit the general tastes, we recruit 20 people to choose preferred images from 18 sets of images. In each testing set, there are two images for each set, one with an evaluation score above 80, and the other one with the evaluation score around 50.

In Figure 49, we show the preferred images from these people in terms of percentage for all 18 sets of images; the blue bars represent high-score images and orange bars represent low-score images. As demonstrated in the results, most testing sets, except sets 1 and 5, show significant preference differences between high-score and low-score images. After interviewing with testers, we find out that some people choose preferred images based on image objects, not based on image compositions. Thus, we believe that our application can assist users to learn how to take photos with better composition techniques.

### 5.3. With/Without our system

In this evaluation, we ask 30 people to take images under 3 different scenes with and without using our application. For the first scene, the compositions of horizontal, rule of thirds, triangle, focus, intensity balance, and contrast are chosen; for the second scene, the compositions of horizontal, vanishing point, frame within a frame, focus, intensity balance, and contrast are chosen; for the third scene, the compositions of central, triangle, vanishing point, frame within a frame, focus, intensity balance, and contrast are chosen.

In Figure 50, we show the average scores of all 30 people under different scenes and with/without using our application. Blue bars represent the scores without using our application while orange bars represent the opposite cases. In the results, we can see that the average scores are in the range from 64 to 72 without using our application, while the average scores are above 80 when using our application. Thus, we believe that our application can improve users’ composition techniques.

## 6. Conclusions and Future Work

We design an application that can evaluate images simultaneously while users take images based on selected composition techniques, and our approach is quite different from most previous works for evaluating images afterwards. Not only geometric compositions are involved, aesthetic elements are also considered in our application. Our application allows users to customize the preferred composition methods and to take pictures automatically when a pre-specified desired composition score is reached. Through the use of our application, users can learn to take images according to some common composition rules.

Due to resource limitations on mobile devices and the concern of execution time, we adopt several techniques with less accuracy. For vanishing point, our application can only detect images with structural roads, and for the frame within a frame, our application performance better for images with a closed frame. Also, we sacrifice image resolution to down-scale images by five times for the frame-within-a-frame composition and by three times for other methods to improve the timing performance. In the future, we expect to deal with images with higher resolutions and adopt better techniques to further improve the performance of our application. Moreover, we hope to consider more composition rules such as symmetry and color harmony, as were considered in [8]. Furthermore, as pointed out in the work by Sachs et al. [29], many existing studies on applying photo composition rules to judge if a photo is good or not tend to “play safe” on photo taking, that is, it is possible that a photo some experts rate as good may not obtain a good score using existing photo ranking systems. How to reduce the gap of judgment between human experts and a software system could be worthy of study. Finally, how to apply machine learning approach to learn a particular photo composition style from another could also be an interesting direction to pursue.

## Figures and Tables

**Figure 1 sensors-20-00582-f001:**
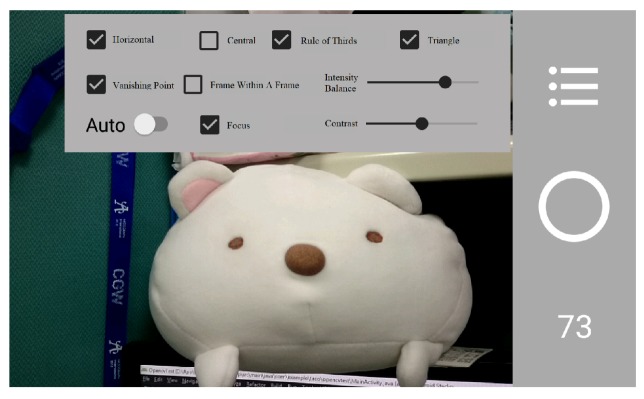
Some interface of our system.

**Figure 2 sensors-20-00582-f002:**
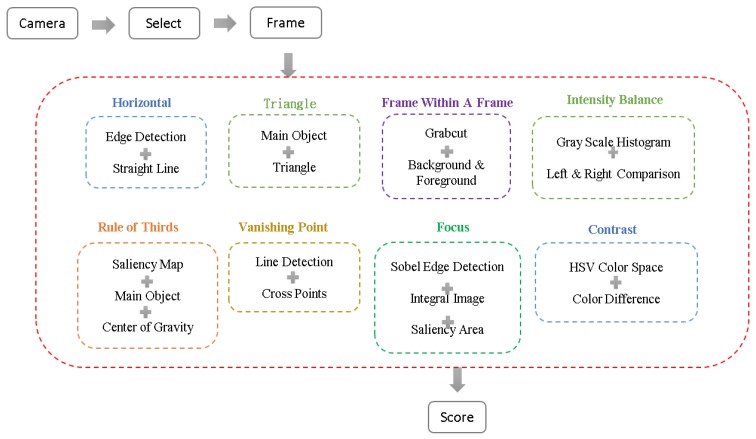
The system flow of this work.

**Figure 3 sensors-20-00582-f003:**
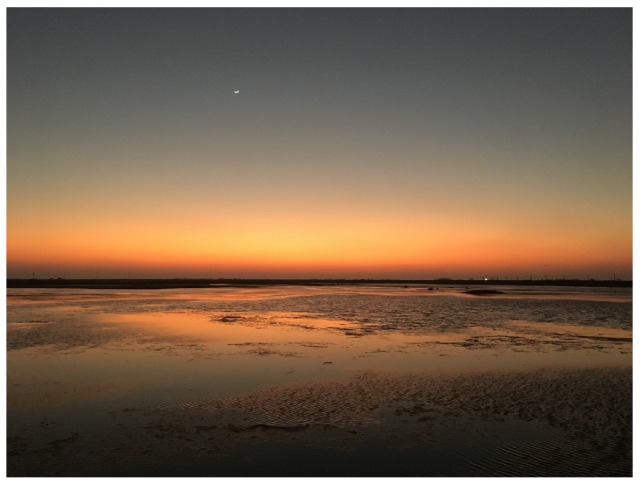
An example of horizontal composition.

**Figure 4 sensors-20-00582-f004:**
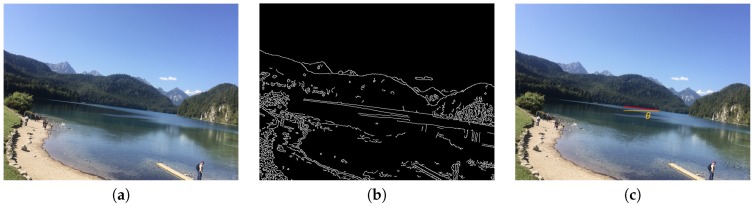
(**a**) Original figure. (**b**) Edge detection results. (**c**) The longest line and its corresponding angle, as shown near the center of the image.

**Figure 5 sensors-20-00582-f005:**
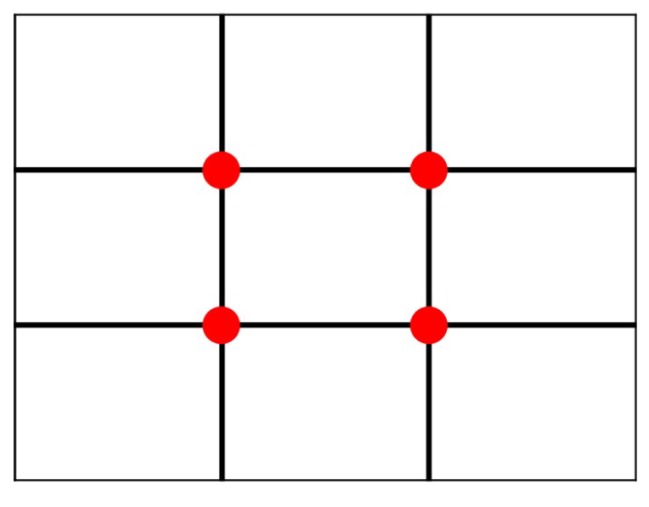
The lines and places (marked in red) where the rule of thirds considers.

**Figure 6 sensors-20-00582-f006:**
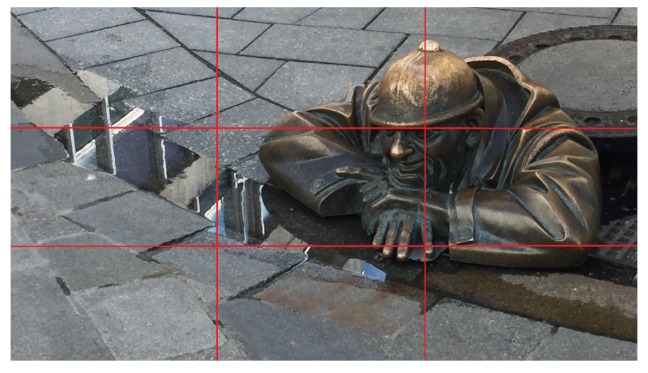
An example image using the rule of thirds.

**Figure 7 sensors-20-00582-f007:**
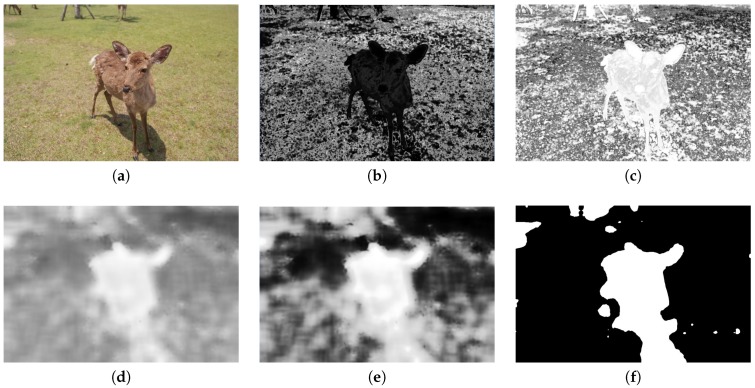
(**a**) Original figure. (**b**) Back-projecting result. (**c**) Reversed result of (**b**). (**d**) Mean shift result of (**c**). (**e**) Result of (**d**) after histogram equalization. (**f**) The saliency region.

**Figure 8 sensors-20-00582-f008:**
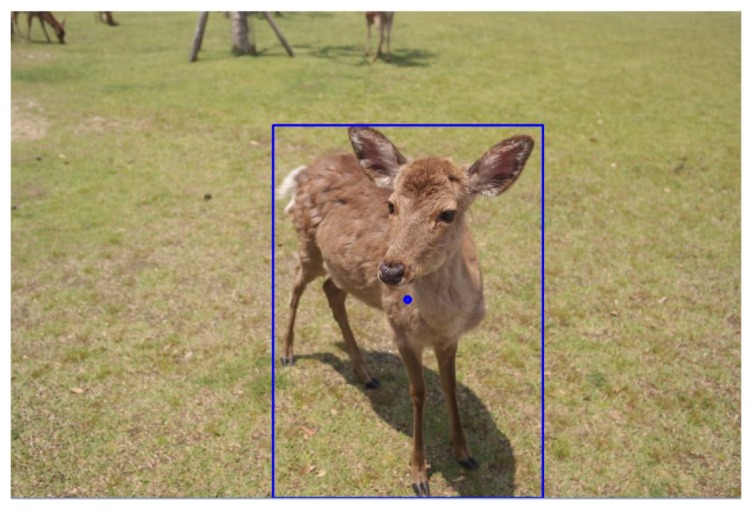
The main body of the foreground and the center of the foreground.

**Figure 9 sensors-20-00582-f009:**
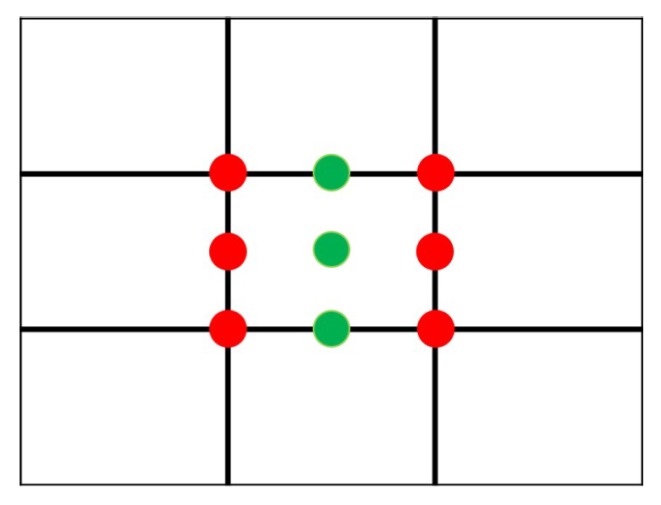
The positions to place the main body of the foreground.

**Figure 10 sensors-20-00582-f010:**
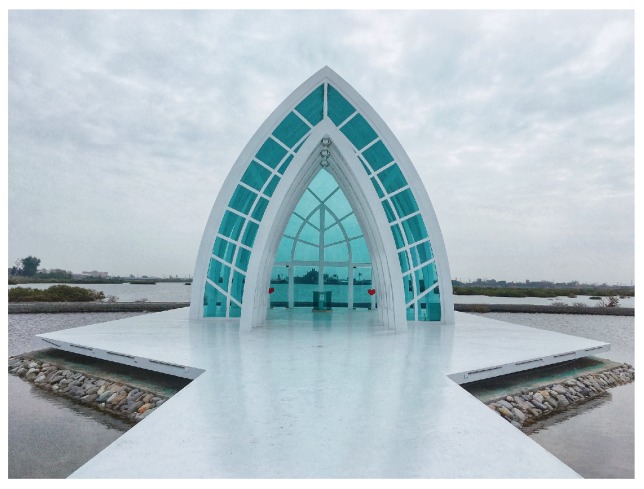
An example image following the triangle composition rule.

**Figure 11 sensors-20-00582-f011:**
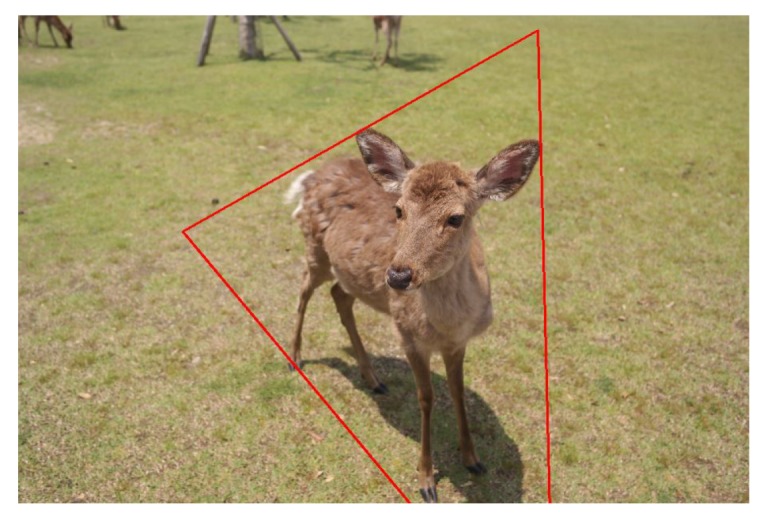
The smallest triangle that can contain the foreground.

**Figure 12 sensors-20-00582-f012:**
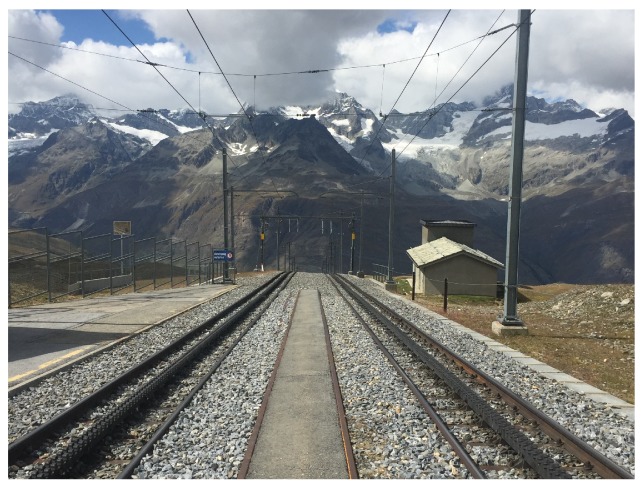
An example image using the vanishing point composition.

**Figure 13 sensors-20-00582-f013:**
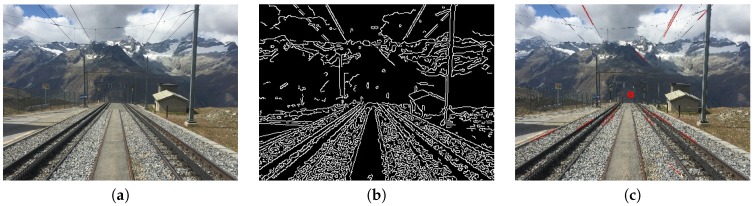
(**a**) Original image. (**b**) Edge detection result. (**c**) Vanishing point calculated result (enlarge to see the red detected lines).

**Figure 14 sensors-20-00582-f014:**
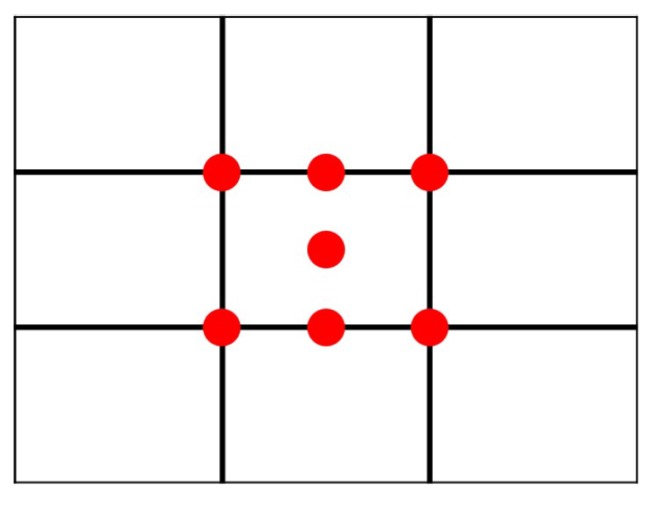
The reference positions to consider a vanishing point.

**Figure 15 sensors-20-00582-f015:**
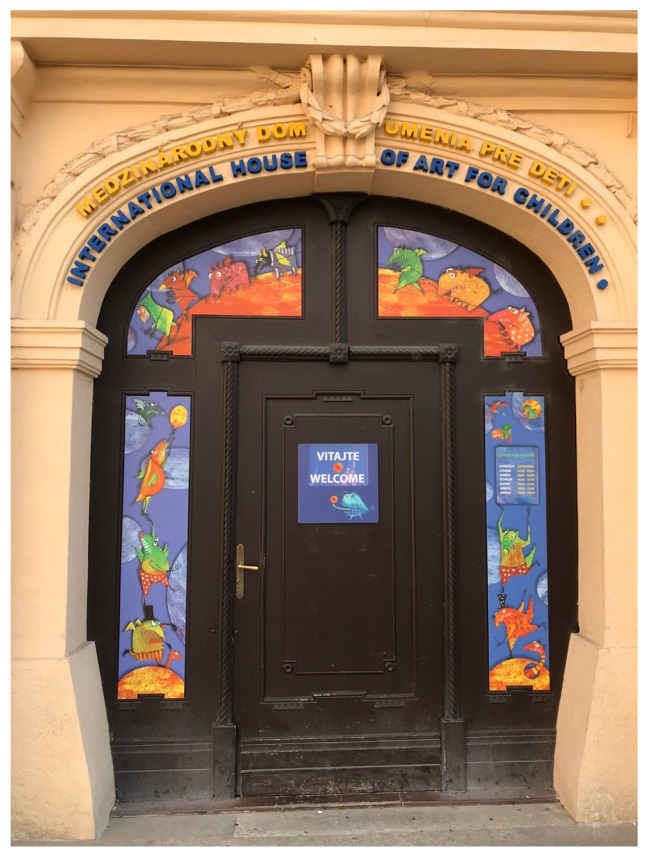
An example using the frame within a frame composition rule.

**Figure 16 sensors-20-00582-f016:**
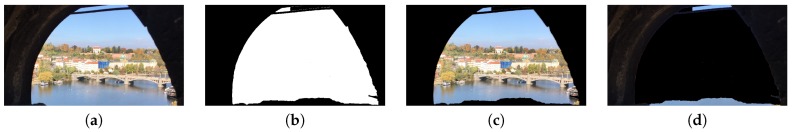
(**a**) Original image. (**b**) The GrabCut mask. (**c**) The obtained background using GrabCut. (**d**) The resulting frame-within-a-frame foreground.

**Figure 17 sensors-20-00582-f017:**
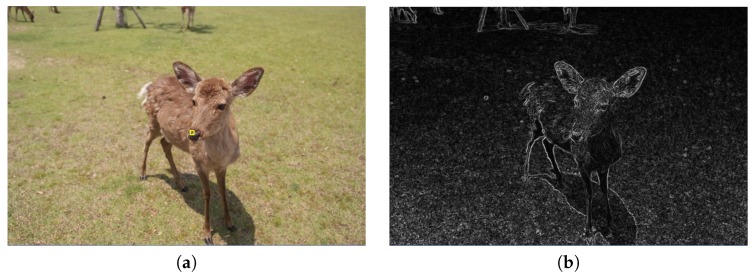
(**a**) Original image. (**b**) Edge detection result.

**Figure 18 sensors-20-00582-f018:**
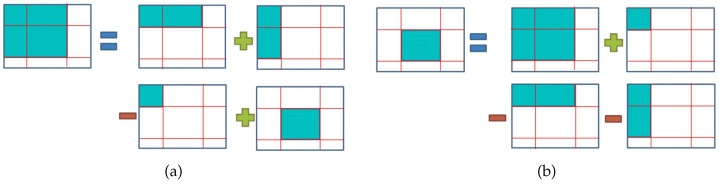
(**a**) The calculation of integral image. (**b**) The way to calculate the total pixel values of a rectangular region.

**Figure 19 sensors-20-00582-f019:**
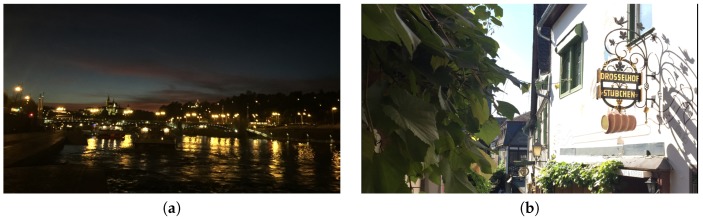
(**a**) An example image with (horizontal) balance of intensity. (**b**) An image without balance of intensity.

**Figure 20 sensors-20-00582-f020:**
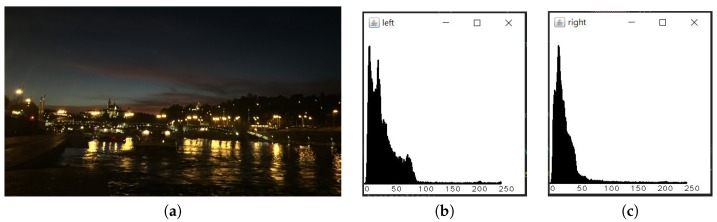
(**a**) Original image. (**b**) The intensity distribution for left half image. (**c**) The intensity distribution for the right half image.

**Figure 21 sensors-20-00582-f021:**
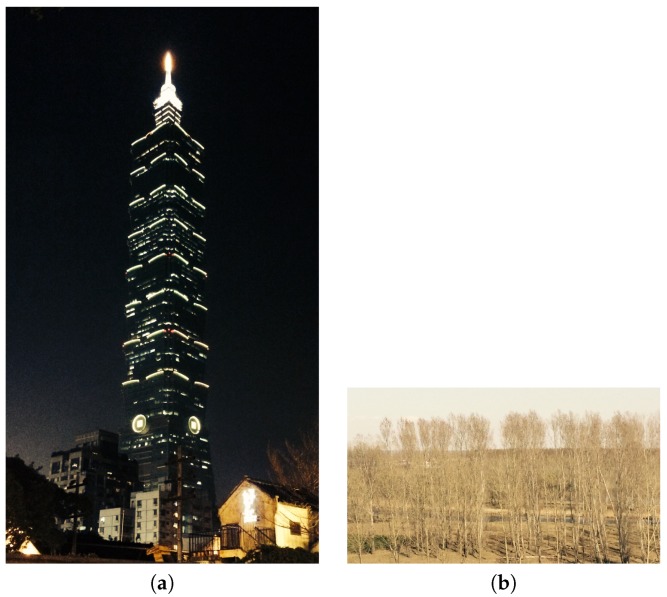
(**a**) An example image with high contrast. (**b**) An image with low contrast.

**Figure 22 sensors-20-00582-f022:**
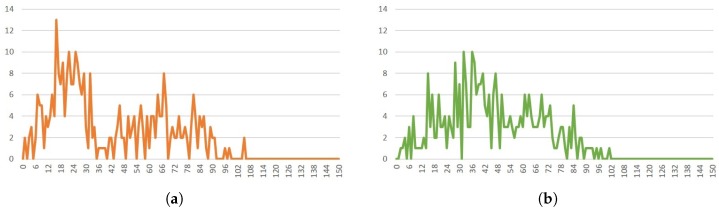
(**a**) The distribution when rule of thirds is considered. (**b**) The distribution for a random photo taking process.

**Figure 23 sensors-20-00582-f023:**
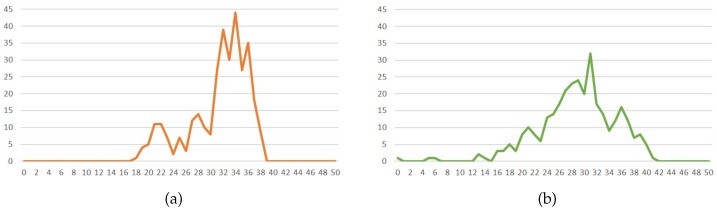
(**a**) The distribution when the focus composition rule is considered. (**b**) The distribution for a random photo taking process.

**Figure 24 sensors-20-00582-f024:**
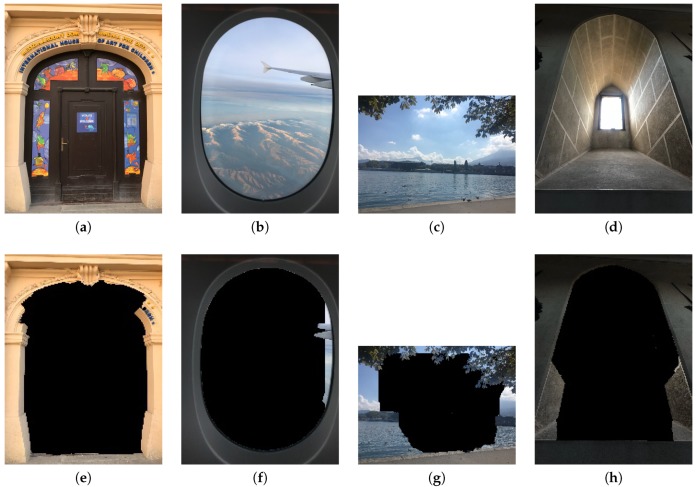
(**a**–**d**) Original images. (**e**–**h**) Foregrounds of (**a**–**d**), respectively.

**Figure 25 sensors-20-00582-f025:**
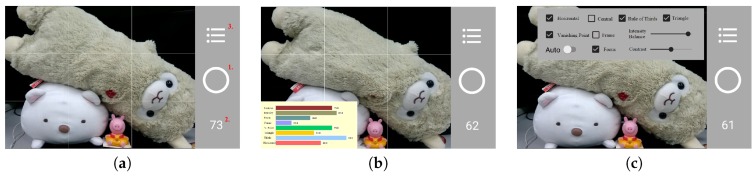
(**a**) Main interface. (**b**) The score for each composition method. (**c**) The interface for setting/selecting composition method (enlarge to see more details).

**Figure 26 sensors-20-00582-f026:**
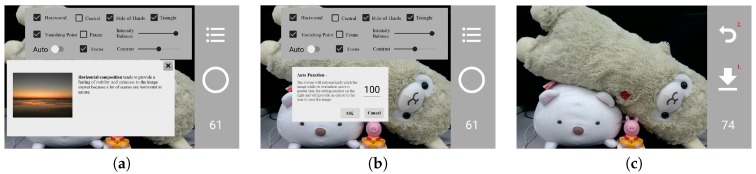
(**a**) Explanation for a composition method. (**b**) The setting of a score value, and when the evaluation score is equal to or greater than this score, the photo will be automatically taken, and the interface will be changed to (**c**).

**Figure 27 sensors-20-00582-f027:**
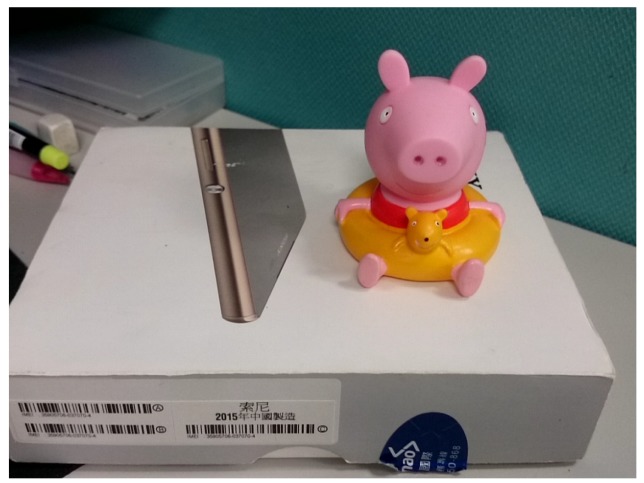
First testing image.

**Figure 28 sensors-20-00582-f028:**
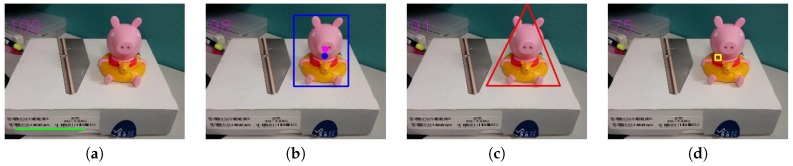
(**a**) Horizontal composition. (**b**) Rule of thirds composition. (**c**) Triangle composition. (**d**) Focus composition.

**Figure 29 sensors-20-00582-f029:**
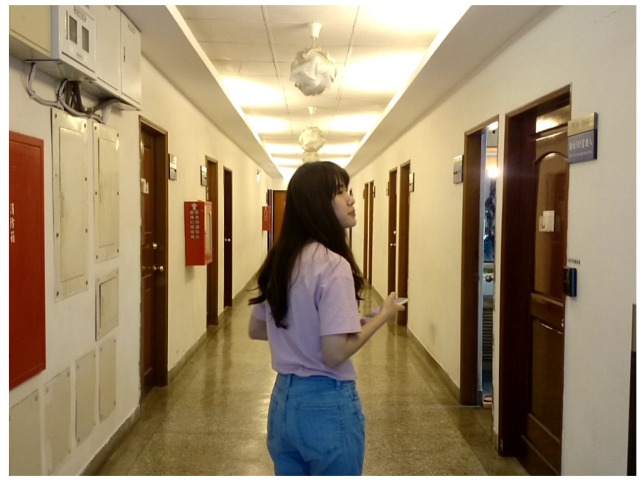
Second testing image.

**Figure 30 sensors-20-00582-f030:**
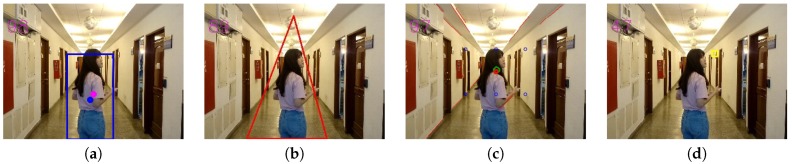
(**a**) Central composition. (**b**) Triangle composition. (**c**) Vanishing point composition (enlarge to see the red detected lines). (**d**) Focus composition (enlarge to see the detected focal point marked in yellow around the upper right region).

**Figure 31 sensors-20-00582-f031:**
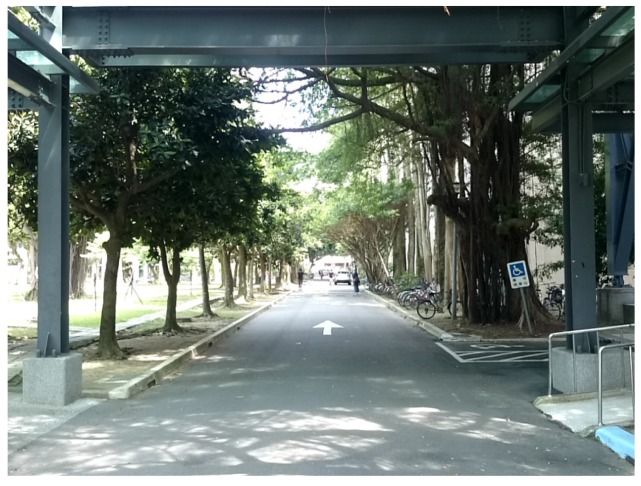
Third testing image.

**Figure 32 sensors-20-00582-f032:**
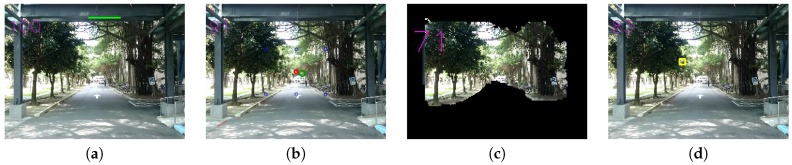
(**a**) Horizontal composition. (**b**) Vanishing point composition. (**c**) Frame within a frame composition. (**d**) Focus composition.

**Figure 33 sensors-20-00582-f033:**
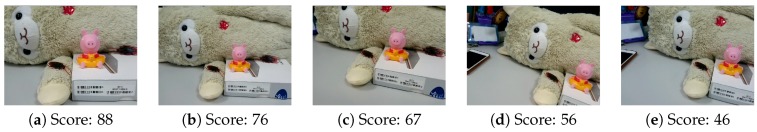
The scores for the first scene using the composition methods of horizontal, rule of thirds, triangle, and focus.

**Figure 34 sensors-20-00582-f034:**
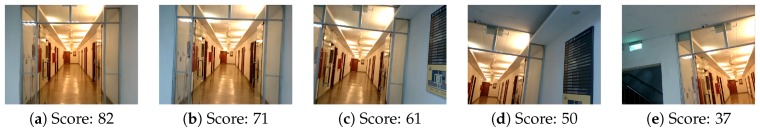
The scores for the second scene using the composition methods of horizontal, vanishing point, frame within a frame, and focus.

**Figure 35 sensors-20-00582-f035:**
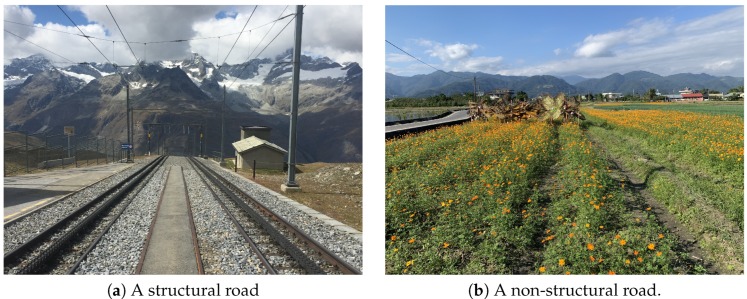
The limitation of finding the vanishing point when the input image is like (**b**).

**Figure 36 sensors-20-00582-f036:**
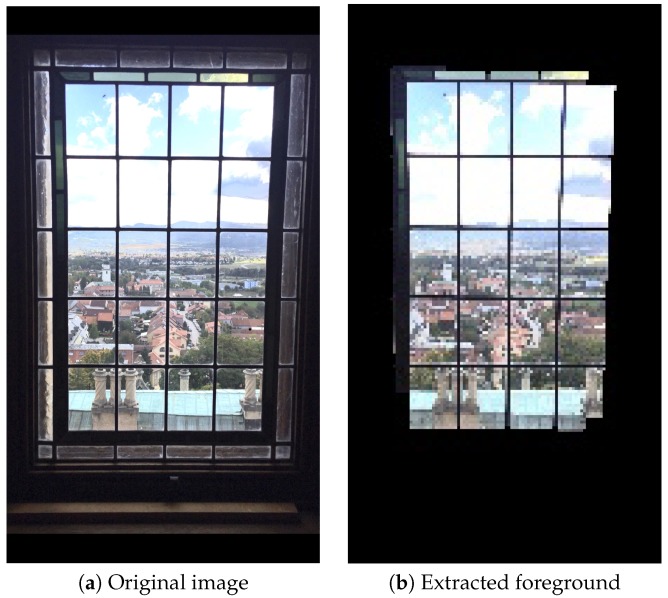
The result of extracting the foreground in a frame-within-a-frame composition method when a closed frame is encountered.

**Figure 37 sensors-20-00582-f037:**
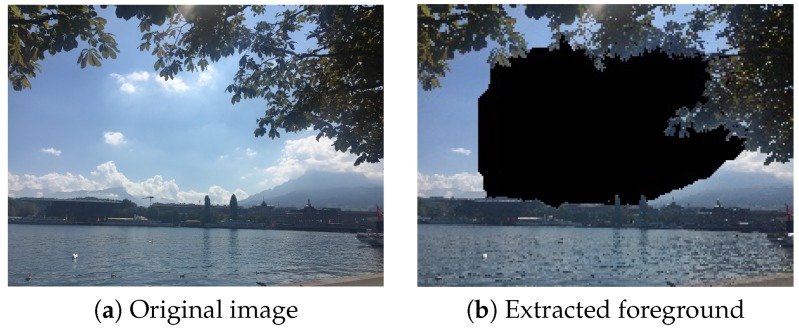
The result of extracting the foreground in a frame within a frame composition method when an open frame is encountered.

**Figure 38 sensors-20-00582-f038:**
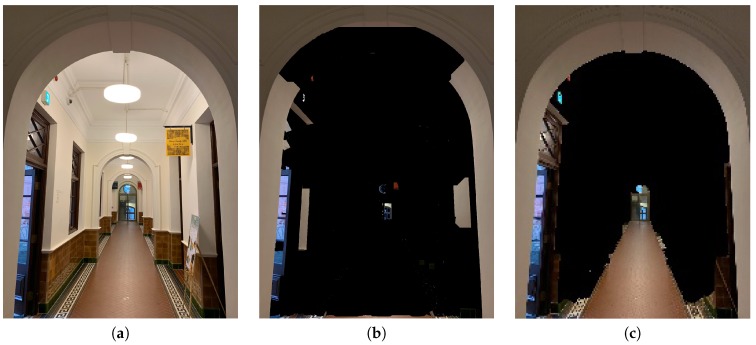
Results of extracting the foreground in a frame-within-a-frame composition method using different resolutions. (**a**) Original image, with the resolution of 1200 × 1600. (**b**) The extracted foreground result using (**a**). (**c**) The extracted foreground result using a reduced resolution of 171 × 228.

**Figure 39 sensors-20-00582-f039:**
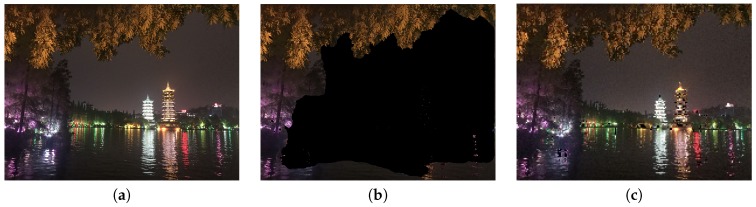
Results of extracting the foreground in a frame-within-a-frame composition method using different resolutions. (**a**) Original image, with the resolution of 4032 × 3024. (**b**) The extracted foreground result using (**a**). (**c**) The extracted foreground result using a reduced resolution of 201 × 151.

**Figure 40 sensors-20-00582-f040:**
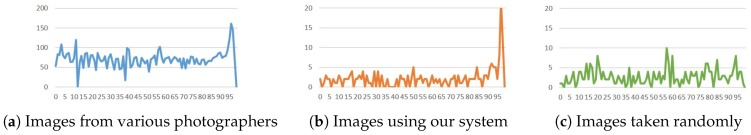
Results for the horizontal composition method.

**Figure 41 sensors-20-00582-f041:**
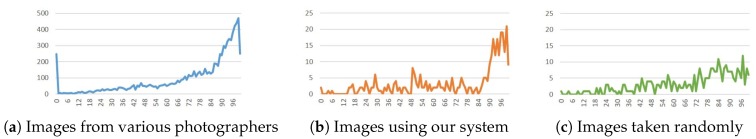
Results for the rule-of-thirds composition method.

**Figure 42 sensors-20-00582-f042:**
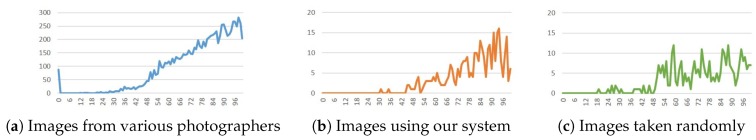
Results for the triangle composition method.

**Figure 43 sensors-20-00582-f043:**
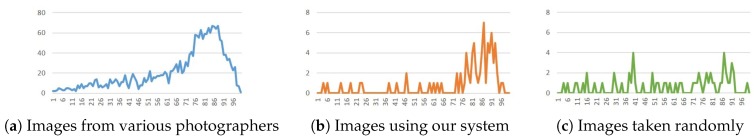
Results for the vanishing-point composition method.

**Figure 44 sensors-20-00582-f044:**
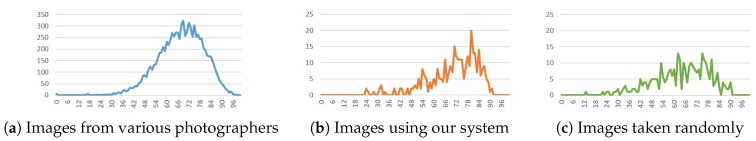
Results for the focus composition method.

**Figure 45 sensors-20-00582-f045:**
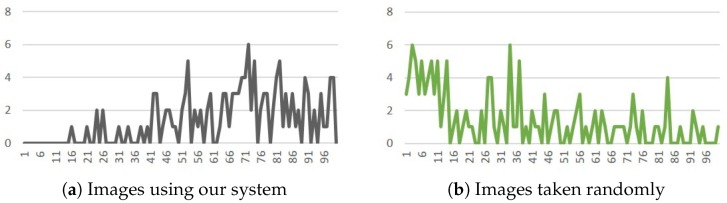
Results for the frame-within-a-frame composition method.

**Figure 46 sensors-20-00582-f046:**
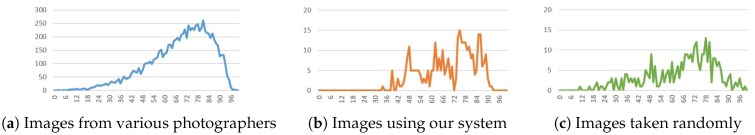
Results for the intensity balance composition method.

**Figure 47 sensors-20-00582-f047:**
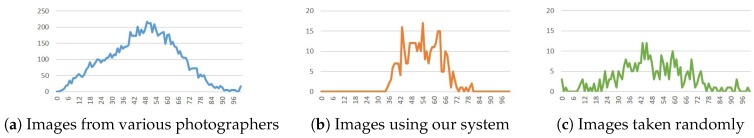
Results for the contrast composition method.

**Figure 48 sensors-20-00582-f048:**
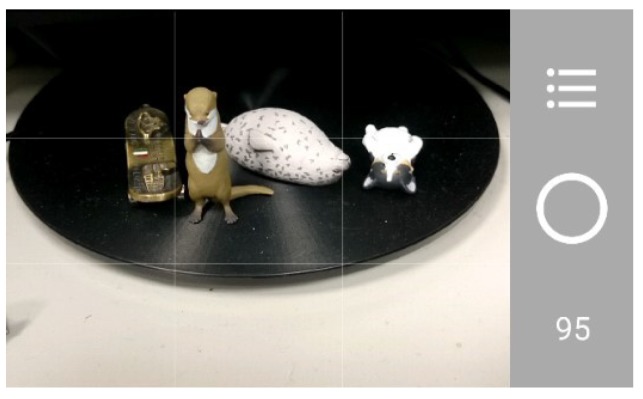
An image with good contrast.

**Figure 49 sensors-20-00582-f049:**
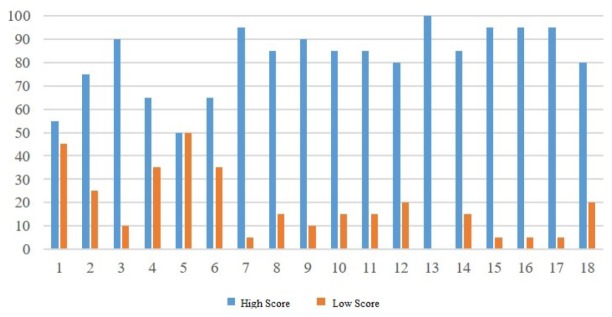
The result of preference.

**Figure 50 sensors-20-00582-f050:**
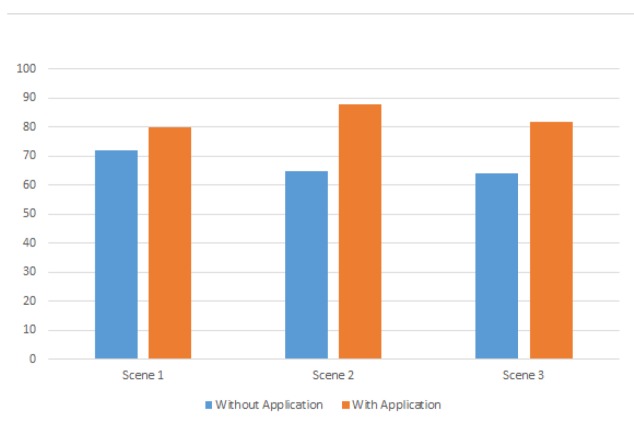
Comparison of average score between with/without using our system.

**Table 1 sensors-20-00582-t001:** The execution time for frame within a frame, in terms of seconds.

Resolution	Time
960 × 720	5.658
320 × 240	0.207
192 × 144	0.073

**Table 2 sensors-20-00582-t002:** System environment.

**Operating System**	Windows 10 64bit
**Processor**	Intel(R) Core(TM)i7-6700 CPU 3.40GHz
**Display card**	NVIDIA GeForce GTX 960
**Memory**	32GB
**Development Platform**	Android Studio, Eclipse
**Mobile Device**	Sony Xperia Z2
**Programming Language**	Java
**Library**	OpenCV-3.4.1, Jama

**Table 3 sensors-20-00582-t003:** The score for each composition method in the first testing image.

Composition Method	Score	Composition Method	Score
horizontal	100	rule of thirds	98
triangle	91	focus	75
intensity (90)	76	contrast (50)	98

**Table 4 sensors-20-00582-t004:** The score for each composition method in the second testing image.

Composition Method	Score	Composition Method	Score
central	98	triangle	89
vanishing	97	focus	47
intensity (90)	91	contrast (50)	95

**Table 5 sensors-20-00582-t005:** The score for each composition method in the third testing image.

Composition Method	Score	Composition Method	Score
Horizontal	100	vanishing	91
Frame within a Frame	71	focus	83
intensity (90)	94	contrast (50)	96

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
