# Peer review of "Photo Composition with Real-Time Rating"

_sensors, 2020, doi:10.3390/s20030582_

Round 1
Reviewer 1 Report
This paper proposed an quality evaluation metric based on several factors. Some experimetns are conducted to show its performance. The paper's content will be useful for some people if the source code is available or the APP is free available. But for researchers, the contents is not so interesting.
The reviewer's suggestions are listed as follows:
how to evaluate the importance of different factors? namely, how to more effectively obtain the weights? how to conduct the subjective evaluation, and more resonable and convinced. it is much better that you can provide the confusion matrix of different composition methods with different evaluation metricsAuthor Response
Please see the attachment.

Reviewer 2 Report
This paper uses several composition rules to evaluate a picture while it is taken. The paper is logically organized and very focused.
Moreover, the reported experimental results seem to be interesting and sufficient detailed.
However, there are a few issues brought to reviewer’s concern:
Line 15 – change lower case p to be upper case. Figure 13c and 30c – hard to see the red lines. Section 3.4.2. Downsampling – How can we know downsampled images will not (or only slightly affect) the processing results (compared with respect to original images)? Line 318 and 322 – change Hugh to Hough. Figure 25c – hard to see the text on the GUI menu. Line 459 – Isn’t 3 accounting only for a very small subset of people? Reference List – not up to date; only contains one paper within the last five years.Author Response
Please see the attachment.

Reviewer 3 Report
Summary:
The paper presents a method and an on-camera application for automatic evaluation of photo composition. The authors propose to evaluate a potential image using eight pre-defined composition rules based on geometrical cues.
Overall, the paper is rather well written, and I appreciate the amount of the work done by authors. However, I believe that the scientific quality of the paper can be largely improved.
The authors address a relevant topic of estimating a photo in the process of acquisition on a smartphone. The proposed method is mostly clearly explained and supported by a number of illustrations. At the same time, the novelty of the proposed methods is limited; some evaluated features resemble minor updates to the features proposed in classic papers on photo aesthetics (for example in the article “Photo and video quality evaluation: Focusing on the subject” by Luo et al.). Also, the use of particular techniques is not always properly motivated or supported by the relevant literature. The results analysis, although supported by a number of graphs, lacks depth and more profound explanations of the observed outcomes. It is also not clear how well the proposed system would perform in comparison with other state-of-the-art solutions.
In my opinion, the paper in its current state cannot be recommended for acceptance. Although it demonstrates a lot of work, I consider that it requires at least a considerable revision. I would recommend authors improve the scientific depth and reasoning of the proposed techniques, along with a clear highlighting of the added novelty. Or, if the authors prefer, they could also highlight the mobile implementation part (since they put considerable efforts in it), but probably it would make it a different submission.
My extra comments are given below:
One of my concerns is that each composition rule is presented without any scientific support or source. For instance, regarding the Triangle composition (line 169): “Triangles are generally regarded as stable shapes and are often used to denote balance. Triangle composition is that the main subjects should portray the shape of a triangle so that it could bring stability, peace and harmony to the image.” No references support this statement and other composition rules. Numerous thresholds without any motivation. In 4.2. a set of thresholds for different techniques is given, and only half of them are explained. Although I understand that pre-defined thresholds are sometimes inevitable, and providing them is useful for reproducibility of results, there should be at least a clarifying sentence about the values. Line 143. “With some mathematical operations”. I suggest avoiding such vague statements. Instead, you better provide the exact equations from Gildenblats’ method, or at least provide a meaningful intuition about these operations. I also encountered similar vague descriptions of mathematical operations in several places within the paper. Also, certain approaches can be potentially improved. For example, a direct histogram difference in 3.2.7 is a very straightforward approach, which is not very robust. I would suggest that the authors check Chi-squared Distance or Earth Movers Distance. The results part presents a lot of graphs, but their usefulness is not obvious, since you mostly compare the scores produced by your method on different photos. These results are difficult to interpret and there are almost no conclusions from these results. There are many examples, but some of them are redundant, e.g. Fig.28 and Fig.30. At the same time, I did not see the test examples where all rules are used together. The user study on with/without using the system is really small (3 users/3 scenes) and cannot be really included in the paper in a current form.
Reviewer 4 Report
TYPOS:
line 15: Photo Compotision (capital letter)
line 139: More (e is missing)
lines 299, 304, 309, 317, 321, 324 (no space before "(" character)
claims:
line 26: "users could experience and learn basic composition techniques and corresponding results to improve their photo taking skills" is not shown or argue.
line 35: "extensive evaluations", disagree, image categories are unbalanced, no input statistical analysis, the aim of the experiments are vague.
Line 64: ListNet is no referenced or explained anywhere.
line 397: "but in a more or less random manner", is not an appropriate phrase for a scientific paper (reformulate)
Comments:
Rule of thirds uses Lab color space first and HSV later. No reason explained or adecuacy.
Calculating the center of gravity, we found the same problem. Euclidean Distance is selected with no reason or alternatives.
Triangle composition uses the isosceles triangle as the highest score, no reason.
Frame within a frame: uses a rectangle area. No possility of using other forms (circle like on a hatch)
Focus: euclidean distance with no reason
weighted combination: weights aren't identified, and even more, it is not specified how are the calculated.
Experiments:
Downsampling from 960x720 to 192x144 but no analysis in the experiments of how that downsampling affects the results.
No experiments with humans. AMT (Amazon Mechanical Turk), which is referenced, could be a good starting point. For example, presenting experimental images to user so that they can evaluate them in terms of "composition" and the study the correlation (Pearson, Spearman, kendall) between users and the presented approach.
Conclusions:
line 476: "users can learn how to take images with good compositions". It is difficult to understand how showing only a number representing the "composition value" can anyone with no more information learn to take better photos. I can only imaging a "trial and error" approach.
References for authors:
Carballal A., Perez R., Santos A., Castro L. (2014) A Complexity Approach for Identifying Aesthetic Composite Landscapes. In: Romero J., McDermott J., Correia J. (eds) Evolutionary and Biologically Inspired Music, Sound, Art and Design. EvoMUSART 2014. Lecture Notes in Computer Science, vol 8601. Springer, Berlin, Heidelberg
Round 2
Reviewer 4 Report
After the changes made I consider the paper ready for publication.